# Variation in Cause-Specific Mortality Rates in Italy during the First Wave of the COVID-19 Pandemic: A Study Based on Nationwide Data

**DOI:** 10.3390/ijerph19020805

**Published:** 2022-01-12

**Authors:** Enrico Grande, Ugo Fedeli, Marilena Pappagallo, Roberta Crialesi, Stefano Marchetti, Giada Minelli, Ivano Iavarone, Luisa Frova, Graziano Onder, Francesco Grippo

**Affiliations:** 1Integrated System for Health, Social Assistance and Welfare, Italian National Institute of Statistics, 00198 Rome, Italy; pappagal@istat.it (M.P.); crialesi@istat.it (R.C.); stmarche@istat.it (S.M.); frova@istat.it (L.F.); frgrippo@istat.it (F.G.); 2Epidemiological Department, Azienda Zero, Veneto Region, 35132 Padova, Italy; ugo.fedeli@azero.veneto.it; 3Statistical Service, Istituto Superiore di Sanità, 00161 Rome, Italy; giada.minelli@iss.it; 4Department of Environment and Health, Istituto Superiore di Sanità, 00161 Rome, Italy; ivano.iavarone@iss.it; 5Department of Cardiovascular, Endocrine-Metabolic Diseases and Ageing, Istituto Superiore di Sanità, 00161 Rome, Italy; graziano.onder@iss.it

**Keywords:** COVID-19, mortality, causes of death, Italy, pandemic

## Abstract

Italy was a country severely hit by the first coronavirus disease 2019 (COVID-19) pandemic wave in early 2020. Mortality studies have focused on the overall excess mortality observed during the pandemic. This paper investigates the cause-specific mortality in Italy from March 2020 to April 2020 and the variation in mortality rates compared with those in 2015–2019 regarding sex, age, and epidemic area. Causes of death were derived from the national cause-of-death register. COVID-19 was the leading cause of death among males and the second leading cause among females. Chronic diseases, such as diabetes and hypertensive, ischemic heart, and cerebrovascular diseases, with decreasing or stable mortality rates in 2015–2019, showed a reversal in the mortality trend. Moreover, mortality due to pneumonia and influenza increased. No increase in neoplasm mortality was observed. Among external causes of death, mortality increased for accidental falls but reduced for transport accidents and suicide. Mortality from causes other than COVID-19 increased similarly in both genders and more at ages 65 years or above. Compared with other areas in Italy, the Lombardy region showed the largest excess in mortality for all leading causes. Underdiagnosis of COVID-19 at the beginning of the pandemic may, to some extent, explain the mortality increase for some causes of death, especially pneumonia and other respiratory diseases.

## 1. Introduction

To date, studies on deaths attributable to coronavirus disease 2019 (COVID-19) have mainly focused on the overall excess of deaths during the pandemic compared with those in previous years [1,2] or on data provided by the surveillance estimating approximately 3,950,832 COVID-19-related deaths worldwide [3]. However, the analysis of causes of death (CoDs) is necessary to understand the direct and indirect effects of the disease on mortality since the pandemic might also affect mortality from many common CoDs other than COVID-19. Increased mortality from preexisting chronic diseases or from acute cardiovascular events [4] might be due to early or late consequences of COVID-19 infection or to social disruption and limited access to care during peak pandemic waves [5]. Nonetheless, few population-based data are available on mortality from multiple disease categories during the pandemic.

Increased weekly mortality from heart diseases, diabetes, and Alzheimer’s disease/dementia has been observed in correspondence with pandemic peaks in the United States [6]. In England and Wales, among confirmed COVID-19 deaths, hypertension, dementia, chronic lung disease, and diabetes were the most frequently reported comorbidities in death certificates, all displaying a higher prevalence than non-COVID-19 deaths [7]. Regarding external CoDs, a decline in mortality from traffic accidents resulting from quarantine measures has been reported in the Suzhou region of China [8]. A huge drop in traffic accident deaths and a slight reduction in suicide mortality were observed in Peru during lockdown [9]. In contrast, COVID-19 might have negatively affected the outcome of low-energy trauma in elderly people, as suggested by the increased 30-day mortality in patients admitted with hip fractures during the pandemic in the UK [10].

Few studies have investigated the distribution of different CoDs in terms of age and gender during the pandemic. There is only evidence that the excess mortality increased with age for both COVID-19- and non-COVID-19-related mortality; however, regarding gender, mortality rates attributed to COVID-19 were higher among males, whereas the excess mortality due to other causes was larger in females [4,11].

Italy was one of the countries more severely hit by the first COVID-19 pandemic wave in early 2020. So far, in Italy, mortality studies have focused on comorbidities reported in small groups of death certificates of patients with COVID-19 [12,13] or on broad categories of CoDs in small geographical areas [14,15]. The availability of data from the national CoD register represents a unique opportunity for a full analysis of the impact of the pandemic on the most common CoDs. This study aims to describe cause-specific mortality in Italy during the first pandemic wave, to investigate the variations in cause-specific mortality rates compared with those in the previous five years in terms of age and sex, and to compare mortality in areas with high COVID-19 incidence with that in areas with low COVID-19 incidence.

## 2. Materials and Methods

Deaths of the Italian resident population that occurred in March and April of 2015–2020 were analyzed. CoDs observed between March 2020 and April 2020, corresponding to the first pandemic wave in Italy, were compared with CoDs observed in the corresponding months of the previous five years. CoDs were derived from the national CoD registry, managed by the Italian National Institute of Statistics, which collects copies of death certificates completed by the medical certifiers for all deaths occurring in Italy. All causes reported on the death certificate are classified according to the International Classification of Diseases, 10th Revision (ICD10) [16], using the semi-automated coding system Iris (www.iris-institute.org, accessed on 3 November 2021), which attributes ICD codes for approximately 80% of cases; the remaining 20% are reviewed by expert personnel. For each case, the underlying CoD (UC) defined by the World Health Organization (WHO) as “the condition that initiated the train of morbid events leading to death” is selected. COVID-19 coding was performed in accordance with the latest recommendations by the WHO [17]. During this analysis, both 2019 and 2020 data collection for the CoD register was provisional. The completeness of the CoD register was assessed by comparing the number of deaths in the CoD register with the overall number reported from the database of daily mortality of the resident population. These latter data, obtained by integrating the National Resident Population Register and the National Tax Register, include deaths of the resident population, regardless of the place of occurrence (in Italy or abroad), and represent a reliable and complete data collection since it includes data from all municipalities. This database is periodically updated, and analyses were performed on the version published on 29 April 2021 [18]. In this study, we referred to this database as the demographic source. By comparing the two sources, at the end of April 2021, the CoD register had received and coded up to 94% of all death certificates of the resident population in the period March 2020 to April 2020 (Appendix A). To account for the incompleteness of data, we estimated the number of deaths by cause by multiplying the cause-specific proportion of deaths from the CoD register by the number of deaths in the demographic source in each stratum of sex, age group, province of residence, and month of death. The details of the calculation procedure are shown in the Appendix A, with a comparison of the estimated number of deaths by cause with the number of deaths reported in the CoD register (Appendix A). Such an estimation procedure was also applied to the 2015–2019 data.

Age-standardized mortality rates for the most common UC were computed using the direct method, using five-year age-group specific rates, except for the 0–29 and 30–49 age groups and the upper age group (100 years and more). For each year, age-specific rates were calculated using the corresponding population on the first January. The Italian population on 1 January 2020, was used as the standard. We calculated age-standardized mortality rates for Italy as a whole, for Lombardy (the largest Italian region, affected by the highest overall mortality during the first pandemic wave), for Northern Italy, excluding Lombardy, and for the rest of Italy. Moreover, standardized mortality rates were calculated by sex and broad age groups (i.e., <50, 50–64, 65–79, and ≥80 years).

The list of ICD10 codes used for defining the CoDs analyzed is shown in Appendix A.

## 3. Results

In Italy, from March 2020 to April 2020, 159,310 deaths occurred. Of these, 29,184 were due to COVID-19 (59% were males and 41% were females) (Table 1). COVID-19 was the leading CoD among males and the second leading CoD (after neoplasms) among females. The mortality rate of COVID-19 for males was more than double that for females (70.9 deaths per 100,000 residents vs 32.7 deaths per 100,000 residents). Compared with the same months of 2015–2019, the all-cause mortality rate increased by 39.0% in males and by 31.9% in females. Remarkable increases in mortality rates were observed for many leading CoDs. Influenza and pneumonia showed the highest increase, with rates rising by 186.3% in males and 159.8% in females. Among other causes, dementia and Alzheimer’s disease (+30.3% in males and +40.4% in females), diabetes (+35.8% in males and +29.3% in females), hypertensive diseases (+30.0% in males and +27.8% in females), and symptoms, signs, and ill-defined diseases (+26.4% in males and +33.3% in females) showed the highest increases. A slight decrease in the mortality rate was observed for neoplasms. Mortality due to external causes (except for accidental falls) was lower than that observed in the previous 5 years, especially for transport accidents and suicide.

The time trends of age-standardized mortality rates by cause from March to April from 2015 to 2020 are shown in Figure 1. The mortality rates for several causes peaked in 2020. Some causes, such as influenza and pneumonia, dementia and Alzheimer’s, and symptoms, signs and ill-defined diseases, showed increasing mortality levels recently; nevertheless, the level reached in 2020 was clearly above the overall recent trend. The mortality rates for diabetes, hypertensive diseases, and accidental falls, which were stable or slightly decreasing from 2015 to 2019, showed a dramatic increase in 2020. Mortality rates for neoplasms continued to decline through 2020, and only a small increase in mortality rates compared with those in recent years occurred for ischemic heart and cerebrovascular diseases. However, a remarkable drop in mortality rates was observed for transport accidents and suicides. Overall, in 2020, mortality for all causes, other than COVID-19, was higher than that observed in previous years.

Regarding 2015–2019, the overall mortality increased in all age groups, with the lowest percent increase (+1.3%) observed in subjects aged less than 50 years (mortality rates for this age group are reported in Appendix A) and the highest increase observed in subjects aged 65–79 years among males (+49.1%) and in subjects aged 80 years or above among females (+35.4%). Table 2 shows age-standardized mortality rates from March 2020 to April 2020 and percent change compared with those from 2015 to 2019 by cause and sex for age groups above 50 years. COVID-19 mortality rates notably increased by age: in the age group of 80 years or above, COVID-19 showed the highest mortality rate with 537.1 deaths per 100,000 residents in males and 302.9 deaths per 100,000 residents in females. Compared with 2015–2019, mortality rates for all causes, excluding COVID-19, slightly decreased in the 50–64 age group (−2.7% in males and −4.0% in females) but increased in the older age groups (65–79 years: +9.1% in males and +6.7% in females; 80 years or more: +12.6% in males and +15.7% in females). The strongest increase in mortality rates from influenza and pneumonia was observed in the 65–79 and 50–64 age groups in both sexes. Mortality rates from diabetes, dementia and Alzheimer’s disease, and hypertensive diseases notably increased at ages 65 years or older in both sexes. A reduction in mortality rates for ischemic heart disease was observed in all age classes among females and in the 50–64 age group among males. Mortality due to transport accidents reduced in all age classes; rates of deaths due to suicide decreased in all age classes, except for ages 80 years or older among males, in which a 10% increase was observed; mortality from accidental falls increased for ages 65 years or older in males and in all age groups in females. For ages less than 50 years (Appendix A), the mortality rate for all causes, excluding COVID-19, was lower than that observed in the previous 5 years (−10% in males and −6.4% in females); mortality for all external causes, especially transport accidents and suicide, remarkably reduced.

In Table 3, age-standardized mortality rates by cause and geographical area of residence are reported. The COVID-19 mortality rate in Lombardy from March 2020 to April 2020 (135.8 per 100,000) was more than double that in other northern regions and approximately 10-fold higher than that in the rest of Italy. Moreover, Lombardy showed the highest increase in mortality rates compared with those observed in 2015–2019 for influenza and pneumonia (percent change: +628.2%), other respiratory diseases, all chronic diseases excluding neoplasms, and symptoms, signs, and ill-defined diseases. Furthermore, mortality rates related to CoDs that did not increase at a national level, such as ischemic heart diseases, showed a large excess in Lombardy. Therefore, in Lombardy, during the first pandemic wave, mortality for all major causes was higher than that observed in the rest of the country, except for transport accidents. The increase in mortality due to influenza and pneumonia, diabetes, dementia and Alzheimer’s disease, and hypertensive disease observed in the rest of Italy (i.e., central and southern regions and islands) was lower than those observed in Lombardy and other northern regions of Italy. Moreover, in these regions, mortality rates for many circulatory diseases and some respiratory diseases were lower than those observed in the previous years.

## 4. Discussion

The analysis of death certificate data performed in this study provided a detailed picture of the huge impact of the severe acute respiratory syndrome coronavirus 2 (SARS-CoV-2) pandemic on cause-specific mortality from March 2020 to April 2020, including age, gender, and territorial differences.

Compared with the recent years, a dramatic rise in mortality for almost all leading CoDs was registered, with the largest increase observed in mortality due to pneumonia and influenza. Some chronic diseases (i.e., diabetes, hypertensive diseases, ischemic heart disease, and cerebrovascular diseases), with decreasing or stable mortality rates from 2015 to 2019, showed a reversal in the mortality trend in 2020. During the first pandemic wave, no increase in mortality was observed for neoplasms, whereas a reduction in mortality rates was observed for transport accidents and suicides. Although mortality due to COVID-19 mainly concerned males, as previously reported [19], we found that compared with those in 2015–2019, mortality from other causes showed a similar pattern of variation in both sexes. As for age, mortality for all causes, other than COVID-19, notably increased for people aged 65 years or older. Regarding geographical differences, compared with other regions of Italy, Lombardy showed the largest excess in mortality for all leading CoDs.

Compared with other countries providing cause-of-death data, an excess of deaths for influenza and pneumonia in the months of March and April was observed in the US [6], with a slight increase in deaths for other diseases of the respiratory system. After this period, no excess in mortality was observed in the US for such causes. This finding suggests that the excess mortality for respiratory diseases, especially influenza and pneumonia, may indicate underreporting of COVID-19, attributable to the difficulty in diagnosing COVID-19 in the first period of the pandemic. In Italy, the countrywide availability of SARS-CoV-2 tests was limited during the March–April pandemic wave, and in the absence of a biological confirmation, deaths may have been attributed to influenza and pneumonia rather than COVID-19.

The rise in mortality from some chronic diseases was also observed in other countries. In the US [6], the weekly trend of excess mortality for circulatory diseases, Alzheimer’s disease, and other dementias showed patterns similar to COVID-19 mortality. In Spain, a slight increase in mortality for nervous system diseases, particularly Alzheimer’s disease, diabetes, and hypertensive diseases was documented in 2020 [20]. In Sweden [21], diseases of the nervous system and behavioral disorders increased during the March–April wave, whereas other causes did not seem to be particularly affected by the pandemic.

The increases in mortality due to non-COVID-19 conditions during the March–April pandemic wave may be due to difficulties in accessing healthcare services by patients with conditions different from COVID-19. Patients may have avoided contact with healthcare services due to the increased risk and fear of contracting COVID-19 [22]. Furthermore, in the lockdown period, healthcare resources were relocated to assist patients with COVID-19, and appointments or visits for conditions different from COVID-19 were often postponed. In some regions, non-urgent healthcare services were canceled. This may have resulted in the undertreatment of relevant chronic and acute conditions, increasing mortality [23]. Moreover, the increase in mortality observed for several leading CoDs may indicate, to some extent, underreporting of COVID-19 [24]. We have mentioned, as in Italy, the lack of biological confirmations of SARS-CoV-2 infection due to the limited availability of tests, which might have led to attributing these deaths to known and documented conditions (i.e., dementia, diabetes, and cardiovascular disease) instead of COVID-19.

In agreement with our findings, no increase in mortality due to neoplasms was observed in other countries. In Italy, subjects with a previous diagnosis of cancer did not have an increased risk of testing positive in a real-time polymerase chain reaction (PCR) for SARS-CoV-2; however, among individuals testing positive, those with cancer were at a higher risk of hospitalization and death [25]. Meanwhile, the number of new first pathological cancer diagnoses decreased during the lockdown, probably due to delays in access to proper diagnostic procedures [26]. In a French multicenter study, among symptomatic patients with cancer, positive on PCR or computed tomography, approximately half of deaths were attributed to COVID-19 according to death certificate data [27]. In view of the aforementioned statements, factors associated with the lack of increased mortality from cancer at the population level during the first pandemic wave might be as follows: patients with cancer were not at a higher risk of infection due to the protective effect of social distancing measures; a proportion of cancer deaths could have been attributed to COVID-19 itself. Lastly, the impact of diagnostic and therapeutic delays on cancer mortality will be better evaluated using data extended through the entire duration of the pandemic.

The drop-in mortality rates for transport accidents agree with what was observed in Peru after lockdown implementation [9]. Italy was the first European country (the second in the world after China) to adopt a hard national lockdown in March and April. As a consequence, the drastic reduction in vehicle circulation had a favorable impact on mortality due to traffic accidents.

The reduction in suicide mortality rates during the first pandemic wave conforms to the findings from other studies [9,28,29]. In England, a decline in hospital presentations for self-harm during the 3 months following the introduction of lockdown restrictions was observed [29]. Different factors can be called upon as possible explanations for the reduction in suicide mortality during the early months of the pandemic. For instance, the lockdown determined closer living circumstances within families, which may have acted as a protective factor for suicide [30], as well as an increased cohesion between individuals in facing the battle against the virus. Moreover, some people could have benefited from a reduction in everyday stress during stay-at-home periods [28]. In Japan [29], after the decline in suicide mortality during the first 5 months of the pandemic, a 16% increase in monthly suicide mortality rates was observed during the second wave (July to October 2020). This result stresses the importance of monitoring suicide rates in the months following the first wave of a pandemic when the full economic and financial consequences of the pandemic are expected to emerge.

The increase in mortality due to accidental falls among older adults is probably attributable to an increase in deaths due to domestic traumas, not preventable by social isolation. Such a hypothesis is consistent with the findings from the UK [10], showing an increase in 30-day mortality in patients with hip fracture during the first months of the pandemic, and Italy [31], showing an increase in deaths among patients with femur fracture in the first eight weeks of the pandemic. The authors of the Italian study have highlighted that COVID-19 infection caused a more complex postoperative course and hospital reorganization, resulting in a significant percentage increase in deaths in the first three postoperative weeks.

According to the study by Gianicolo et al. [32], the excess mortality was observed in all age groups; mortality for all causes, other than COVID-19, was observed only for people aged 65 and more. This might suggest that COVID-19 has a smaller indirect effect on younger people and a greater protective effect due to the containment of traffic accidents. Nevertheless, in the younger age groups, especially in males, there was an excess in mortality from pneumonia and other respiratory diseases, suggesting an underestimation of COVID-19 cases.

This study adds to sparse reports on the larger excess of non-COVID-19-related mortality in females during the pandemic [4,11]. Notably, Kontopantelis et al. have found larger excess mortality in females from a residual category of causes different from COVID-19 and respiratory diseases, diabetes, cardiovascular diseases, and cancer. Consistent with these findings, we found that the increase in overall mortality was significantly larger in males than that in females; however, when COVID-19 is excluded, the increase was larger in females, although the statistical difference between sexes was less evident (Appendix A). Table 1 shows that the percent changes between 2020 and 2015–2019 were higher in females, especially those for dementia and Alzheimer’s disease, cerebrovascular diseases, other respiratory diseases, and ill-defined diseases; for the latter two disease categories, this might be due to the underreporting of COVID-19.

In the first wave of the pandemic, the scenario in Italy was characterized by a strong and rapid increase in cases and deaths and by a high geographical concentration, mainly in the northern part of Italy [33]. The Lombardy region was the most severely hit area in Italy, with high COVID-19 mortality rates. Moreover, mortality from non-COVID-19 CoDs increased to a much larger extent than that in other geographical areas: for many diseases, mortality rates from March 2020 to April 2020 were double those observed in the reference period. The only exceptions were the decline in mortality due to transport accidents and suicide and cancer mortality being roughly unchanged. The impact of the underdiagnosis of COVID-19 infection on non-COVID-19-related deaths is difficult to disentangle from the effects of social disruption and difficulties in access to care, especially in areas where the pandemic initially overwhelmed the diagnostic capacity of the healthcare system. Note that only in Lombardy, an increase in mortality from ischemic heart disease was registered during the first epidemic wave. A decline in hospital admissions for myocardial infarction was registered during the lockdown in Italy; patients with minor symptoms probably avoided or delayed access to the Emergency Department due to social isolation and fear of contagion within hospitals [34]. Such further barriers to timely care, possibly increasing out-of-hospital cardiovascular mortality and worsening prognosis of hospitalized patients, were likely stronger in areas with higher COVID-19 incidence.

Although regional and local differences in the healthcare services are documented in Italy, it is difficult to assess their role in the excess mortality of 2020, especially during the first pandemic wave. One of the reasons is that the outbreak in March–April involved mainly the North of the country, especially three regions (Lombardy, Veneto, and Piedmont) with highly performing health care services and with very similar indicators of efficiency and effectiveness. In addition, the Lombardy region was the first European area to be affected, and the spread of the pandemic was so fast that it overwhelmed every health care system.

Air pollution might explain the observed geographical pattern of the increase in mortality for COVID-19 and other CoDs in Italy in the first pandemic wave with respect to 2015–2019. A large body of evidence on the impact of air pollution on human health has accumulated over the past few years [35], and the SARS-CoV-2 outbreak caused a fast proliferation of new epidemiological studies linking ambient air pollution to COVID-19. Though air pollution increases the risk of chronic diseases (e.g., respiratory and cardio-metabolic diseases), the same ones described as the comorbidities increasing the risk of being hospitalized or dying from COVID-19 [36], the contribution of chronic exposure to air pollutants to modulating the spread and severity of COVID-19 remains unclear. This is mainly due to the design of available studies (most were ecological and correlation studies), with low resolution (data aggregated at the regional or province level), and their inability to control for community and territorial contextual factors [36,37]. Recent studies conducted in Italy have suggested that short- and long-term exposure to air pollutants contributes to modulating the spread and severity of COVID-19 [38,39]; however, the aforementioned shortcomings limit the interpretation of their findings. These limitations are being addressed in an ongoing Italian epidemiological study program (EpiCovAir), coordinated by the National Institute of Health in collaboration with the National System for the Environmental Protection. EpiCovAir is analyzing the association of long-term air pollution (PM_10_, PM_2.5_, and NO_2_) with the incidence of and mortality from COVID-19 in Italy at the municipal level; however, this study can account for municipal- and individual-based covariates, looking separately at epidemic waves and geographical patterns.

The main limitation of this study is that the data analyzed are restricted to the first epidemic wave. Further analyses are warranted to assess the impact of the late consequences of COVID-19 infection on causes of mortality, the CoDs pattern in subsequent pandemic waves, and to investigate possible long-term effects of limited access to care through 2020 among patients affected by chronic diseases. An additional limitation is that the applied method uses the information on CoDs collected by the CoD register to estimate the number of deaths by cause in the resident population obtained from the demographic source. The estimation was performed considering age, sex, and a fine territorial subdivision (provinces). This procedure allowed us to overcome the slight under-coverage of CoD data due to delays in data transmission by the different territories. Nevertheless, the estimates were based on 94% of observed deaths, and the percentage of under-coverages was homogeneous by region and age group (Appendix A). The estimation procedure only resized the number of observed deaths by cause based on the total number of deaths in the demographic source. Thus, no biases in the contribution of each CoD were introduced (Appendix A).

## 5. Conclusions

In conclusion, the analysis of cause-of-death data allowed us to deeply investigate the excess mortality in the first wave of the pandemic and to evaluate both the direct and indirect effects that have affected the Italian territories. The underdiagnosis of COVID-19 at the beginning of the pandemic may, to some extent, explain the increase in mortality observed for some CoDs, especially pneumonia and other respiratory diseases. However, the increase in mortality for some chronic conditions or accidental falls might be related to the indirect effects of the pandemic. Future analyses extended to the entire 2020 and subsequent years are needed to assess the long-term impact of the COVID-19 pandemic on CoDs.

## Figures and Tables

**Figure 1 ijerph-19-00805-f001:**
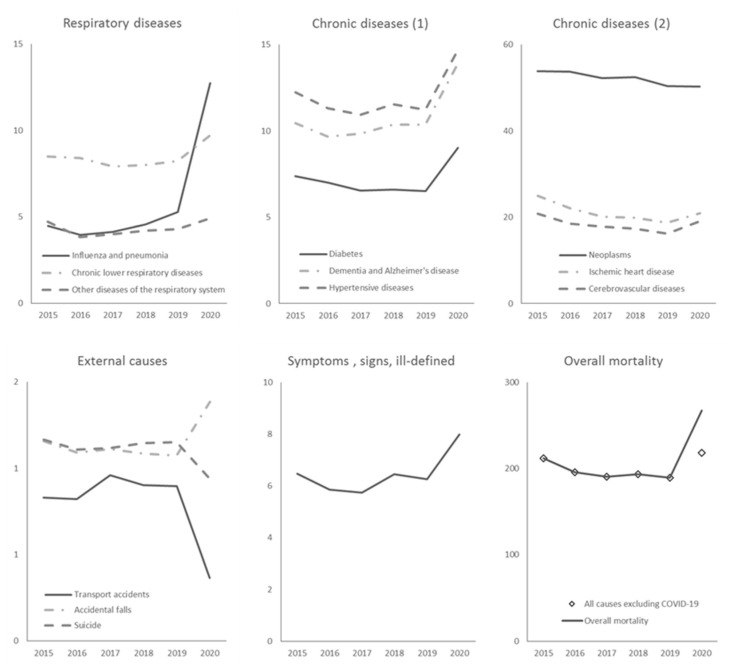
Time trend of age-standardized rates by cause (per 100,000 residents) from March to April for Italy from 2015 to 2020.

**Table 1 ijerph-19-00805-t001:** Mortality by cause in Italy from March 2020 to April 2020 for all ages: estimated number of deaths, age-standardized mortality rates (per 100,000 residents), and percent change in rates compared with respect those in 2015–2019.

	Males			Females			Males and Females		
	No. of deaths 2020	Rates		No. of deaths 2020	Rates		No. of deaths 2020	Rates	
Causes of death		2020	Percent change 2020/2015–2019		2020	Percent change 2020/2015–2019		2020	Percent change 2020/2015–2019
COVID-19	17,160	70.9		12,024	32.7		29,184	48.9	
Infectious and parasitic diseases (excluding COVID-19)	1144	4.8	−3.3	1262	3.4	−9.0	2405	4.0	−5.7
Neoplasms	16,587	66.7	−5.7	13,386	39.1	−2.9	29,973	50.3	−4.3
Diabetes	2495	10.7	35.8	2881	7.7	29.3	5376	9.0	32.6
Dementia and Alzheimer’s disease	2551	12.0	30.3	5768	14.7	40.4	8319	13.9	37.4
Hypertensive diseases	3041	14.9	30.0	5735	14.4	27.8	8776	14.7	28.5
Ischemic heart disease	6561	29.4	1.1	5908	15.2	−3.5	12,469	20.9	−0.9
Cerebrovascular diseases	4477	20.3	2.6	6928	18.0	7.4	11,404	19.1	5.8
Other diseases of the circulatory system	5068	23.6	1.0	7064	18.1	2.4	12,133	20.3	2.2
Influenza and pneumonia	4072	17.7	186.3	3533	9.3	159.8	7605	12.8	183.7
Chronic lower respiratory diseases	3240	15.0	16.5	2569	6.7	18.8	5809	9.7	18.7
Other diseases of the respiratory system	1405	6.4	11.5	1534	4.0	22.8	2939	4.9	17.3
Symptoms, signs, and ill-defined diseases	1854	8.7	26.4	2912	7.3	33.3	4766	8.0	30.1
Transport accidents	175	0.6	−57.7	44	0.1	−60.5	219	0.4	−58.5
Accidental falls	444	2.0	25.4	382	1.0	26.1	826	1.4	25.3
Suicide	454	1.7	−13.2	106	0.3	−29.9	560	0.9	−17.5
Other external causes	1020	4.6	−4.2	1242	3.2	−0.8	2263	3.8	−2.9
Other causes	6833	29.0	11.5	7452	20.3	8.6	14,285	24.0	10.7
All causes excluding COVID-19	61,421	268.2	9.9	68,705	183.0	11.9	130,126	218.2	11.4
All causes	78,581	339.1	39.0	80,729	215.7	31.9	159,310	267.1	36.3

**Table 2 ijerph-19-00805-t002:** Mortality by cause in Italy from March 2020 to April 2020 by sex and age class (50 years and above): age-standardized mortality rates (per 100,000 residents) and percent change in rates compared with those in 2015–2019.

	Age Class (Years)					
	50–64		65–79		≥80	
Causes of death	Rate	Percent change 2020/2015–2019	Rate	Percent change 2020/2015–2019	Rate	Percent change 2020/2015–2019
(Males)						
COVID-19	25.9		156.5		537.1	
Neoplasms	36.3	−9.1	157.2	−7.8	436.2	−3.2
Diabetes	3.1	23.0	18.4	28.1	95.9	41.4
Dementia and Alzheimer’s disease	0.3	2.2	10.7	34.8	139.2	29.9
Hypertensive diseases	2.0	1.6	14.1	40.0	163.8	29.9
Ischemic heart disease	8.3	−16.9	42.9	0.9	276.8	3.6
Other diseases of the circulatory system	5.0	3.4	26.3	−3,5	243.2	2.0
Cerebrovascular diseases	3.6	13.6	25.4	10.4	207.0	0.2
Influenza and pneumonia	3.5	315.5	30.8	442.1	160.5	135.3
Chronic lower respiratory diseases	1.5	2.3	17.3	11.0	160.6	18.1
Other diseases of the respiratory system	1.3	11.6	8.1	4.5	62.8	12.3
Symptoms, signs, and ill-defined diseases	3.5	6.7	8.6	24.6	82.8	33.4
Transport accidents	0.6	−61.1	0.7	−62.6	1.8	−51.4
Accidental falls	0.4	−22.2	2.8	36.0	19.2	33.1
Suicide	1.7	−30.0	2.3	−18.1	4.9	10.1
All causes, excluding COVID-19	86.7	−2.7	427.3	9.1	2375.9	12.6
All causes	112.7	26.4	583.7	49.1	2913.0	38.1
(Females)						
COVID-19	6.7		53.7		302.9	
Neoplasms	28.1	−8.0	91.1	−2.2	224.5	−1.9
Diabetes	1.3	24.3	10.3	20.8	77.6	32.5
Dementia and Alzheimer’s disease	0.2	−19.5	10.1	31.3	177.0	42.0
Hypertensive diseases	0.6	−6.4	9.0	18.4	173.1	29.3
Ischemic heart disease	1.9	−5.5	14.0	−12.9	169.1	−1.9
Cerebrovascular diseases	1.9	2.9	17.5	13.9	198.8	6.2
Other diseases of the circulatory system	1.5	−31.3	15.7	−3.6	205.0	4.5
Influenza and pneumonia	1.2	185.3	11.1	270.3	98.2	142.4
Chronic lower respiratory diseases	0.7	−6.2	7.7	11.5	72.1	21.8
Other diseases of the respiratory system	0.7	38.1	4.6	21.3	42.1	22.9
Symptoms, signs, and ill-defined diseases	1.3	23.8	4.5	17.7	82.7	37.4
Transport accidents	0.1	−67.0	0.1	−77.6	0.2	−72.4
Accidental falls	0.2	81.5	1.0	22.6	10.8	26.7
Suicide	0.5	−34.2	0.5	−28.1	0.5	−20.0
All causes, excluding COVID-19	48.0	−4.0	234.5	6.7	1782.8	15.7
All causes	54.7	9.5	288.2	31.1	2085.7	35.4

**Table 3 ijerph-19-00805-t003:** Mortality by cause in Italy from March 2020 to April 2020, by geographical area of residence: age-standardized mortality rates (values per 100,000 residents) and percent change in rates compared with those in 2015–2019.

	Lombardy		Northern Italy (Excluding Lombardy)		Rest of Italy	
Causes of death	Rate 2020	Percent change 2020/2015–2019	Rate 2020	Percent change 2020/2015–2019	Rate 2020	Percent change 2020/2015–2019
COVID-19	135.8		59.1		14.4	
Infectious and parasitic diseases (excluding COVID-19)	5.7	28.2	4.3	−11.0	3.3	−14.2
Neoplasms	55.0	−0.8	50.2	−3.8	48.7	−5.8
Diabetes	10.2	140.2	7.3	40.1	9.7	12.9
Dementia and Alzheimer’s disease	26.9	145.9	13.1	22.9	10.3	7.5
Hypertensive diseases	17.3	109.7	13.1	30.9	14.9	11.1
Ischemic heart disease	25.2	38.0	18.7	-0.8	20.9	−10.7
Cerebrovascular diseases	24.1	49.0	18.1	10.5	18.1	−8.2
Other diseases of the circulatory system	24.8	37.6	20.4	5.7	18.9	−9.7
Influenza and pneumonia	37.1	628.2	12.8	136.3	4.8	30.3
Chronic lower respiratory diseases	15.1	114.6	7.7	8.3	9.3	0.1
Other diseases of the respiratory system	9.3	151.8	4.4	9.0	3.8	−14.0
Symptoms, signs, and ill-defined diseases	11.2	134.0	7.3	41.2	7.4	2.5
Transport accidents	0.3	−59.7	0.3	−66.0	0.4	−53.2
Accidental falls	1.7	68.8	1.6	21.1	1.1	14.2
Suicide	1.0	−21.0	1.2	−11.7	0.8	−20.6
Other external causes	4.2	34.3	2.9	−4.6	4.2	−10.2
Other causes	30.5	52.2	24.5	12.8	21.5	−2.6
All causes, excluding COVID-19	299.6	64.1	208.0	10.8	198.0	−3.5
All causes	435.4	138.4	267.0	42.3	212.4	3.5

## Data Availability

Relevant data for calculating the indicators presented in this study are publicly available on the website of the Italian National Institute of Statistics (www.istat.it, accessed on 3 November 2021), particularly in the section “causes of death” of the following page: https://www.istat.it/it/archivio/240401 (accessed on 3 November 2021). Links to the data used in this study are also shown in the reference list.

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
