# Peer review of "Variation in Cause-Specific Mortality Rates in Italy during the First Wave of the COVID-19 Pandemic: A Study Based on Nationwide Data"

_ijerph, 2022, doi:10.3390/ijerph19020805_

Round 1

Reviewer 1 Report

Manuscript by Grande et al. examines the variations in mortality in Italy during the first wave of Covid-19, comparing national data with those of a previous 5-year period, with the aim of identifying significant variations in different regional areas of Italy. The rationale is quite interesting and the data help to better understand the dynamics occurred during the pandemic events related to Covid-19. In general, the manuscript appears well written, sufficiently concise and clear, although I personally think that some aspects should be considered more carefully. The first question concerns the state of the health system at the beginning of the pandemic events: any data related to the state of efficiency, effectiveness, readiness of the health system response should be considered if available. Are there any differences, from this point of view, on a regional or local basis? Could these differences have led to an excess of mortality due to Covid-19 and / or consequently to other causes? I believe these aspects are of crucial importance and probably the assessments and statistics proposed here should be integrated with such data and evidence. The second aspect is related to the fact that actually there are several works in the literature focusing on the interaction of the environment with the diffusion and severity of Covid-19: as regards Italy, for example, it has been observed that the different rates of chronic atmospheric contamination at the regional level may have contributed to modulating the spread and severity of Covid-19 (see for example manuscript by Fattorini and Regoli 2020 study, Environ Pollut 264: 114732). European Community demonstrate how atmospheric contamination affects mortality rates every year, therefore one wonders if and how certain environmental variables can mutually affect mortality due to Covid-19 and other causes. Again, such topics should be addressed, and relevant literature cited. Finally, I think that the manuscript can be considered acceptable if the aspects described above will be implemented.

Reviewer 2 Report

This work is relevant, and has a special characteristic: it is also a starting point for more studies in the area and its publication may very well challenge other researchers to do so, including other regions or even countries. The problem is a rapidly-changing one, and evaluations of this kind are fundamentally important.

In line 75, even though I am not a native english speaker it seems to me that the correct form could be:

"which attributes ICD codes for about 80% of cases; the remaining 20% is are reviewed by...", since the subject of attributes is singular - "semi-automated coding system Iris" - and for the "are", the subject is 20% which seem plural. 

But this does not, in any way affect the comprehension of the text. And, from my point of view, it important to publicize this work and so my reccomendation is that it should be accepted forr publicarion.

Reviewer 3 Report

Thank you for inviting me to review the study entitled “Variation in cause-specific mortality rates in Italy during the first wave of COVID-19 pandemic: a study based on nationwide data (ijerph-1480789)” related by Grande et al. 

This manuscript reports mortality rates in Italy between the years of 2015 to 2020, considering COVID-19 and other causes of death. It is an interesting study that evaluated the association between gender and age with the number of deaths in the beginning of the first wave in that country. The main aim of this study was to verify if COVID-19 contributed to the increases of mortality rate associated with the variables mentioned above. 

However, this study presents some considerations to improve its quality and to clarify its idea. 

First, the background was written based on the types of deaths that have occured during the pandemic in other countries. Even though the causes of death was demonstrated, the aim of the study was proposed to investigate the difference between causes of death, COVID-19 mortality associated to age and sex. The introduction did not mention any of those variables. Thus, a robust introduction was missed by the authors and remains required.

The main concern is the methodological aspects of the study.

Was the estimated number of deaths calculated by a validated instrument?

The classification of age groups does not correspond to those presented in the tables. In these aspects, the contradictory writing is a limitation of interpretation by the readers. (Please see methods, Table S2, Table 2).

In addition, the number of causes of death for COVID-19 shown in the Table S3 (27,650) is different from the reported in the results (29,184). 

Authors included the cardiovascular diseases mortality as a single group (Table S3), but its demonstration in Table 1 is different (Hypertensive diseases; Ischemic heart disease; Cerebrovascular diseases; and other diseases of the circulatory system). Data from other diseases of the circulatory system  were not reported in Table 2.

Even though the all-cause mortality rate has increased between male and female (39% vs 31.9%). It seems that COVID-19 has a role in this situation, when this  variable was excluded a slight difference was observed between female (11.9) vs male (9.9). Is there a statistical difference between these variables?

Figure 1 (B, C) presents mortality rates lower and higher than 15 per 100,000, respectively. However this classification must be described in the methods. Why were these parameters considered by the authors?

Most importantly, no statistical analysis was performed.

Authors have focused on discussing the causes of mortality, but they did not related any aspect related to age and sex. Both variables were the main aim of the study. This manner presented by the author is inappropriate and contradictory, considering the objective mentioned in the manuscript. 

The authors need a native English-speaking person to revise the grammar of this manuscript.

Round 2

Reviewer 1 Report

Personally I believe that the authors' responses to the first review can be considered exhaustive, therefore the manuscript can be considered acceptable. Best regards.

Reviewer 3 Report

Although the authors had improved the manuscript quality, reporting guidelines provided by Enhancing the QUAlity and Transparency Of health Research (Equator network) were not related. By considering that an observational study requires a STROBE checklist to ensure a manuscript can be understood by a reader and replicated by a researcher, and also the absence of better information about limitations and biases, I maintain my decision of rejecting the paper.

In addition, I reported in the first reviewing process that the authors have focused on discussing the causes of mortality, but they did not related any aspect related to age and sex. In the new version of the manuscript, only one paragraph was added to discuss the questions asked before that compromise the quality of the article. 

Author Response

Although the authors had improved the manuscript quality, reporting guidelines provided by Enhancing the QUAlity and Transparency Of health Research (Equator network) were not related. By considering that an observational study requires a STROBE checklist to ensure a manuscript can be understood by a reader and replicated by a researcher, and also the absence of better information about limitations and biases, I maintain my decision of rejecting the paper.

Re: We believe that our paper meets the requirements of quality reporting of the results and transparency of the research.

Background and rationale of the investigation are reported in the introduction, and the aims of the study (focus on cause-specific mortality) are clearly stated.

The methodology is presented in detail in the methods section including the description of the population under study, the reference period of data, data coverage, sources of data, data collection and coding of death certificates, the estimation procedure, calculation of mortality indicators, and causes of death and subgroups of the population for which the indicators were provided. In addition, further details on data completeness and the estimation procedure are provided in the supplementary material including a comparison of total number of deaths from the two sources considered, a comparison between the estimated number of deaths and the observed number of deaths by the cause-of-death Register, and the full list of ICD-10 codes used for calculating mortality indicators by cause.

All data are used for the analysis are published online or available upon request.

The main results of the investigation have been illustrated and the key findings with reference to the study aims have been summarized and discussed. Limits and potential biases of the study have been extensively  mentioned and discussed in the last paragraph before the Conclusions section.

The easy calculation procedure and the clear indication of ICD-10 codes used to compute mortality indicators allow for the replication of the study in each country with available routinely collected data on causes of death and resident population.

In addition, I reported in the first reviewing process that the authors have focused on discussing the causes of mortality, but they did not related any aspect related to age and sex. In the new version of the manuscript, only one paragraph was added to discuss the questions asked before that compromise the quality of the article.

Re: The focus of the paper is on causes of death, as clearly stated in the introduction and in the aims. Our effort was mainly devoted to discuss the variation in mortality by cause occurred in the first pandemic wave, since only few studies have focused on this topic. Conversely, aspects related to mortality by sex and age in the first pandemic wave have been already addressed in other studies, so we have chosen to limit the discussion of findings related to these aspects. However, we point out that in the discussion section (page 9) there are two paragraphs in which we discuss separately findings related to age and sex.